# Software-Defined Vehicular Cloud Networks: Architecture, Applications and Virtual Machine Migration

**DOI:** 10.3390/s20041092

**Published:** 2020-02-17

**Authors:** Lionel Nkenyereye, Lewis Nkenyereye, Bayu Adhi Tama, Alavalapati Goutham Reddy, JaeSeung Song

**Affiliations:** 1Department of Computer and Information Security, Sejong University, Seoul 05006, Korea; lionelnk82@gmail.com (L.N.); nkenyele@sejong.ac.kr (L.N.); 2Department of Mechanical Engineering, Pohang University of Science and Technology, Pohang 37673, Korea; btama@acm.org; 3Department of Computer Science and Engineering, National Institute of Technology, Andhra Pradesh 534101, India; goutham@nitandhra.ac.in

**Keywords:** cloud-based vehicular network, virtualization, cloud computing, software-defined vehicular cloud network, multi-access edge cloud, vehicular cloud, virtual machine migration

## Abstract

Cloud computing supports many unprecedented cloud-based vehicular applications. To improve connectivity and bandwidth through programmable networking architectures, Software- Defined (SD) Vehicular Network (SDVN) is introduced. SDVN architecture enables vehicles to be equipped with SDN OpenFlow switch on which the routing rules are updated from a SDN OpenFlow controller. From SDVN, new vehicular architectures are introduced, for instance SD Vehicular Cloud (SDVC). In SDVC, vehicles are SDN devices that host virtualization technology for enabling deployment of cloud-based vehicular applications. In addition, the migration of Virtual Machines (VM) over SDVC challenges the performance of cloud-based vehicular applications due the highly mobility of vehicles. However, the current literature that discusses VM migration in SDVC is very limited. In this paper, we first analyze the evolution of computation and networking technologies of SDVC with a focus on its architecture within the cloud-based vehicular environment. Then, we discuss the potential cloud-based vehicular applications assisted by the SDVC along with its ability to manage several VM migration scenarios. Lastly, we provide a detailed comparison of existing frameworks in SDVC that integrate the VM migration approach and different emulators or simulators network used to evaluate VM frameworks’ use cases.

## 1. Introduction

Vehicular applications include safety, geo-advertising, and entertainment applications to make the vehicle’s passengers journey more enjoyable. The European Transport and Telecommunication Information (ETSI) defines Cooperative-Intelligent Transport System (C-ITS) applications that provide emergency (safety) messages and reduce drastically car accidents [1]. In fact, C-ITS applications include, for instance, in-vehicle speed limit or emergency collision. Messages from C-ITS applications can be generated from different use cases such as an accident and are generally only one or two hops. In addition, geo-advertising constitutes a set of applications for advertising to vehicles within a specific zone area and direction. Furthermore, entertainment-based vehicular applications are mainly cloud-based applications which improve safety, comfort, and entertainment through the development of technology allowing for Internet equipped vehicles referred to as the Internet of Vehicles (IoV) [2]. IoV offers passengers the ability to access cloud resources anytime and pay as they use them. Therefore, the cloud computing helps to continuously perform other activities such as editing meeting presentation files or attending an important team meeting as long as the vehicle is connected to the Internet. However, the high mobility and frequent handover from serving Road Side Units (RSUs) increase network overload due to the number of re-connection requests and insufficient vehicle computing resources. Moreover, the issue of backhaul latency affects Quality of service (QoS) since many cloud-based vehicular applications are bandwidth-hungry. Thus, cloud computing provides considerable resources to enable cloud-based applications including IoV applications.

The Vehicular Cloud (VC) architecture [3,4] is in the progress of merging with the IoV as a fundamental platform for the Intelligent Transportation System (ITS). The cloud architecture for vehicular networks consists of three interacting layers: VC, RSU cloud, and central cloud [4]. Furthermore, the components of central and RSU cloud are described as a highly virtualized platform that provides the computation, storage, and network resources. In addition, Edge Computing (EC) is an extension of a central cloud to provide less delay and better computation performance [5]. An example of a similar system to EC, the cloudlet, is defined as cloud computing in s box [6]. Cloudlets are deployed at the RSU as a small-scale operational site that offers cloud services to the vehicles in the immediate vicinity of the cloudlets. These cloud services are generally initiated from central/edge cloud infrastructures and migrated to the vehicles through mobility management protocols. However, the dynamic topology of the vehicular networks and high mobility diminish data dissemination, which leads to the implementation of complex routing protocols. Thus, complex routing protocols limit the implementation of new vehicular applications, which would allow everything around the vehicle to communicate with the vehicles. Therefore, emergent network technologies were recently proposed to address existing issues in vehicular networks. Presently, the Software-Defined Network (SDN) is among the emergent network technologies for Vehicular Ad-hoc NETworks (VANETs) [7] and has drawn much attention from many academic researchers.

The SDN is a growing networking concept in VANETs that brings better performance through Software-Defined (SD) Vehicular Network (SDVN). In SDVN, the vehicles and radio access at the RSU are considered as SDN devices (data plane). The SDN controller generates and maintains a database that includes network-wide knowledge such as network topology [8,9]. In addition, the SDN concept together with EC is proposed as a prominent solution to existing challenges in vehicular networks. Both hierarchical and centralized SDVN architectures improve rapid packet delivery, low latency, and overhead communication [10,11,12,13]. Furthermore, SDVN integrated with network slicing, artificial neural network, and 5G technologies improves both on-demand routing protocols and bandwidth utilization [14,15,16]. The communication efficiency is improved using SDVN integrated 5G with edge computing [17,18]. SDVN provides better performance of location awareness services through Cloud-Radio Access Network (C-RAN) and Mobile Edge Computing (MEC) [19,20]. In brief, the architecture of SDVN improves resource utilization, selection of best routes, and facilitates network programming for next-generation vehicular networks.

SDVN, Virtual Machine (VM) replication, and migration are joined together to improve the bandwidth of cloud-based vehicular applications. SDN-based Vehicular Cloud (SVC/SDVC) [21,22] uses VM service to enable the distribution of software updates on vehicles based on SDN and cloud computing. SDVC architecture considers vehicles to host virtualization services that enable a VM to migrate from one RSU to another. In fact, the SDN controller is responsible for VM migration and replication. Furthermore, existing works on SDVN-based cloud computing integrate EC to connect information providers and information consumers on an individual level [9]. In fact, individual levels of information providers and consumers mainly focus on resource management, loss of connectivity, and placement of SDN controller and their sub-SDN controller to enhance connectivity [9]. Therefore, none of these works discuss the migration of vehicle’s VM service in SDVC and its related applications. Although the concept of SDN can meet the improved demands of VANETs such as high throughput, low communication latency, high mobility, routing overhead, etc., the current available literature that discusses the migration of VMs over SDVC is limited. To the best of our knowledge, this paper is the first to discuss how VM would be managed in SDVC.

In this paper, we discuss the working methodology of the state-of-the-art VM migration over SDN-enabled vehicular cloud computing integrated with MEC and VC. We also investigate the SDN-based vehicular cloud architecture by providing an extensive discussion of its functional system components to control the VM migration approach, mobility prediction, and heterogeneity of wireless interfaces equipped in vehicles. Then, we present the assistance of control layer to manage cloud-based vehicular applications on the basis of their ability to share computation, storage and bandwidth resources. Finally, an array of VM strategies and other design based VM migration techniques that require the attention of the researchers are presented. The design of VM migration modules to be executed into SDN controllers would allow predictable control of MEC forwarding devices during VM migration in SDVC. The main contributions of this article are as follows:Firstly, we provide a comprehensive literature review on network and computation technologies for supporting SDN-based Vehicular Cloud (SDVC) such as VM migration, SDN, vehicular cloud computing, and cellular-to-everything (C-V2X) communication mode.Secondly, we provide an extensive description of SDVC in which the network forwarding devices such as vehicles and RSUs are decoupled from the control plane. The SDVC’s control plane is abstracted and managed by application functions available at either the edge or central cloud infrastructure.Thirdly, we outline and discuss potential cloud-based vehicular applications assisted by the SDVC. We further describe the ability of SDVC’s network control plane to manags several VM migration scenarios due to the high vehicle mobility in VC.Lastly, we provide a detailed comparison of existing frameworks in SDVC that integrate the VM migration approach, VM strategies, and different emulator/simulator networks used to evaluate VM frameworks’ use cases.

The remainder of the paper is organized as follow. The background of technologies behind SDVC is detailed in Section 2. An architecture of SDVC and a description of SDVC’s control plane for assisting potential cloud-based vehicular applications are discussed in Section 3. The management migration of VM over SDVC and a description of existing framework for supporting VM migration are detailed in Section 4. We conclude our work in Section 5.

## 2. Background of Technologies behind SDVC

In this section, a comprehensive review of the related work on the evolution of computing and network technologies for SDVC is presented. The section introduces: firstly, the VC; secondly, the vehicle-to-everything communication technologies; thirdly, the VM service in VC; and, finally, SDN and its concept for vehicular networks.

### 2.1. Vehicular Cloud Computing

Figure 1 shows a high level of cloud-based vehicular application networks. This architecture consists of three interacting layers: VC, MEC cloud, and central cloud. Vehicles are mobile nodes that leverage cloud resources and services.
Vehicular cloud: A local cloud that clusters a certain number of neighboring vehicles. An inter-vehicle network in an ad-hoc manner is established by Vehicle-to-Any (V2X) communications. The vehicles and everything in direct communication are considered as mobile cloud sites. They cooperatively create a vehicular cloud service that distributes computing and networking resources among the participating vehicles [23].MEC cloud: A edge cloud established among a set of distributed multi-access edge cloud. The MEC cloud provides a provision of various V2X-based cloud services through the distribution and integration of distributed edge cloud servers and cloud radio access networks according to the location of V2X users [24,25].Central cloud: The traditional central cloud computing, which is established among a group of commodity servers on the Internet [4]. A V2X user accesses a central cloud through Vehicle-to- Network (V2N) or cellular network using C-V2X communication [23].

Cloud-based vehicular applications and context-aware applications are offered to passengers in vehicles through cloud computing. To address the requirements for a proper Quality of Experience (QoE), the Vehicular Cloud Computing (VCC) thus opens up considerable new opportunities to execute context-aware applications without using resources of the traditional cloud infrastructure [26]. VCC aims at providing cloud-based vehicular services through a dynamic, self-organized cloud formation based on the radio access of neighboring vehicles [26]. Similar to the traditional cloud, VCC can support services by prioritizing computation, storage, and network resource sharing.

Resources sharing in VCC enables computation, storage, and bandwidth as a service because of insufficient vehicle resources. In fact, sharing computing resources as a service, vehicles can offload compute-intensive tasks to the MEC cloud located at the proximity of the willing (consumer) vehicles. Indeed, cloud-based vehicular services are deployed as a VM. The replication and migration are managed either at the central cloud or at the edge cloud. On the other hand, VCC provides cloud-based vehicular service that requires shared storage as a service. Thus, a distributed storage at different edge computing infrastructures is a solution, e.g., for real-time on-board diagnostic (OBD) data [27]. Therefore, the vehicle does not have to store a huge amount of data. A distributed storage implements the storage abstraction so that at any time the collection of distributed data storage would be uploaded to the central cloud at a convenient interval for further analysis. A shared bandwidth service enables vehicles to participate cooperatively in executing high-bandwidth cloud services initiated by a vehicle inside a VC. For instance, downloading a huge volume file would take less time than making one vehicle to complete the download alone.

### 2.2. Vehicle-to-Everything Communication

The Third Generation Partnership Project (3GPP) through mobile broadband standardization has published LTE-V2X technology as a new standard for deploying V2X applications [7]. The proposed LTE-V2X technology is a derivative of the cellular uplink technology that maintains similarity with the current LTE systems. In fact, the LTE-V2X includes two working modes: cellular communication (LTE-Uu) and direct communication (PC5) [23]. The LTE-Uu connection uses the traditional licensed cellular network to implement Vehicle-to-Vehicle (V2V) and vehicle-to-pedestrian (V2P) communication by forwarding (as shown in Figure 2). The LTE-Uu uploads V2X messages to the application server through the stationary User Equipment (UE)-type RSU. Moreover, PC5 enables direct communication between two vehicles through a Device-to-Device (D2D) protocol. In fact, the D2D communication protocol maintains direct communication between nearby UE-capable devices. Currently, D2D protocol enables direct communication for up to 1000 devices in proximity range of 500 m. Thus, D2D devices operate in peer-to-peer network topology mode [23]. In fact, the RSUs forward V2X messages in the target area by establishing direct communication between vehicles over PC5 (as shown in Figure 2). Therefore, the advantages of D2D are the technical aspects of direct device-to-device discovery and battery life efficiency inter-operable with each device [23].

### 2.3. Virtual Machine Service in Vehicular Cloud Computing Network

A MEC cloud is accessible only by the nearby vehicles, i.e. those located within the radio coverage area. This fact helps us recall the concept of an RSU cloudlet [4]. An RSU cloudlet refers to a small-scale RSU cloud site that offers RSU service to the closest vehicles. A vehicle can select a nearby RSU cloudlet and customize a transient RSU service for use. Here, we consider the customized RSU service as a transient RSU service because the RSU can only serve the vehicle for a while [4,6].

Figure 3 shows the dynamic VM synthesis approach to deliver a VM state to infrastructure [6]. A vehicle selects a nearby MEC server. The latter customizes a transitory cloud service to be used for a while by the vehicle. This transitory cloud service is offered as virtual resources in terms of VM. The vehicles are considered mobile nodes that have a VM, named VM-overlay, from the MEC infrastructure that already possesses the VM-base. The VM at the MEC server is named VM-base [4]. A VM-base is a resource template recording the basic structure of a VM, while a VM-overlay mainly keeps the concrete resource requirements of the customized VM [4]. The customization of of VM to serve the vehicle is explained as follows. The vehicle sends the VM-overlay to the MEC server that already possesses the VM-base. The MEC server reloads the VM-base and applies the VM-overlay to VM-base. Finally, the MEC server completes the customization to deliver the dedicated VM. The MEC server executes the dedicated VM to allow the vehicle to use it to run transient cloud service. During the cloud service, as the vehicle moves along the RSU, it switches between different MEC infrastructures. During the handover to the next MEC infrastructure, the vehicle needs to wait for a short time before the next MEC server resumes the dedicated VM. This process of migrating the dedicated VM to the next MEC is referred to as VM migration. During the re-initiation time of the cloud service, the dedicated VM might disconnect temporarily. In the case the vehicle drives out of radio coverage of MEC infrastructure, only a minimum delay will be experienced as long as the vehicle remains out of the radio communication range. In addition, the issue of loss of connectivity closes the Transport Communication Protocol (TCP) established with the next MEC infrastructure; consequently, the next MEC server would not resume the dedicated VM. After the vehicle completes the cloud service running through the dedicated VM, it alerts the MEC server, which in turn discards the VM.

MEC computing should be able to keep cloud service by leveraging VM migration strategy. VM migration decision-making based enables MEC computation resources to keep the dedicated VM one hope distance from their owners (vehicles). These VM migration techniques reduce latencies and downtime when the VM resume at the next MEC infrastructure. For instance, Teka et al. proposed a VM migration strategy based on Multi-Path TCP (MPTCP) approach to keep the VM hosts using the same Internet Protocol (IP) to follow the movements of the users [28]. Establishing a MPTCP [29] is a method for performing a single Internet protocol between communication nodes in wireless communication. The authors of [28] used the MPTCP transport layer to enable the connection to remain established after a Mobile Device (MD) changes its IP address during handover. MPTCP reduces significantly the delay. However, MPTCP technology cannot solve the issue of a single IP address except advanced VM techniques such as live migration of the dedicated VM or Markov decision process-based service migration [30].

### 2.4. Software-Defined Network and Its Concept for Vehicular Networks

The basic logical structure of an SDN is shown in Figure 4. It consists of three planes: the application plane, the control plane, and the forwarding plane. The control plane includes an SDN controller cluster that is deployed as an independent device. The SDN controller communicates the control decisions to the forwarding plane (networking devices). Furthermore, the SDN controller also retrieves information from those network devices to make proper routing decisions. To enable communication with the forwarding devices, SDN uses protocols named SDN control protocol. The SDN control protocol is categorized in northbound and southbound protocols. The SDN controller interacts with the application plane using the northbound interface. The northbound protocols support application developers to manage the network through a program. Besides the northbound protocol, the southbound protocols enable communication from the control plane to the data plane. In addition, network devices based on OpenFlow protocol [31] accept user-defined policies once installed to allow the routing algorithms to take place. The application plane communicates available actions by instructing new paths to the control plane. The control plane carries selected routing paths to update flow tables of data plane. Since the SDN puts the intelligence of the network into a central software-based control, programs and scripts that automatically react to expected and unexpected events can be built directly into the SDN controller.

Communication between the control plane and forwarding plane in the SDN environment takes place through the SDN control protocol. In the SDN, because the control plane and forwarding plane are decoupled, a standard protocol with multivendor support was needed for communication between them. OpenFlow was developed for this purpose. OpenFlow was the first communication protocol that gives access to SDN controllers to program the rest of forwarding network devices [32]. OpenFlow maintains the flow table [31] on the device, which contains actions to be followed for each incoming packet [31]. The SDN controller can then use OpenFlow to program the forwarding plane of an OpenFlow-enabled switch by altering its flow table. Furthermore, the OpenFlow works in two modes to populate the flow tables: the reactive and proactive modes. The reactive mode is the default method of executing control signaling messages and assumes that OpenFlow does not have any prior knowledge of the control plane running on the network devices. In this mode, the first incoming packet of the network traffic received at any of the forwarding devices is forwarded to the SDN controller. Next, the SDN controller uses this information to generate new rules across the whole network. This creates the flow table on all subsequent forwarding devices in the path, and then forwards the network traffic according to new rules implemented in the Openflow switch of the forwarding devices. On the other hand, the proactive mode requires the SDN controller to be pre-configured with some default flow table values. In simple terms, the traffic flow is programmed preemptively as soon as the OpenFlow switch is rebooting. In addition, a secure channel, such as Secure Socket Layer (SSL) or Transport Layer Security (TLS), is used by the OpenFlow protocol to secure the communication between the SDN controller and OpenFlow switches [31].

The SDN is an imperative technology to address existing challenges in vehicular networks. The control logic for existing networks is integrated into every underlying On-board Unit (OBU) [33] hardware equipped in vehicles. The configuration of new routing algorithms is notably difficult because nearly every in-vehicle communication interface needs to be set up separately. Therefore, the solution is to extend the existing VCC by integrating SDN technology [9]. SDN suggests separation of data plane from the control plane with well-defined programmable interfaces to provide a centralized global view of vehicular networks. SDN defines an easier way to configure and manage vehicular networks. In the same manner, a centralized system has better consistency since the SDN controller has a global view of the network. SDN has evolved in design of next generation of vehicle networks [33].

The SDVN pioneers the concept of SDN in vehicular networks [33]. The control plane is responsible for collecting and maintaining the network management of all SDN enabled radio access at RSUs and vehicles. An example of an application based on SDVN is the prediction of optimal driving paths on demand. The application layer in SDVN architecture could monitor vehicles on the roads, and provide additional best navigation paths at certain times of the day or when the vehicles are temporarily disconnected due to the high road traffic.

The ability to deploy an important number of vehicular applications through SDVN concepts brings the true meaning of the Intelligent Transportation System (ITS). Today, deployment of ITS applications demands higher agility in network restoration, massive scalability, faster deployment, and operating expenses optimization. ITS services simply cannot afford to be slowed down by the lack of speed in human-driven processes. Automation and programmable capabilities are needed to support the provisioning of ITS services, the monitoring of ITS networks, implementation of run-time changes due to the high mobility of vehicular networks and road traffic loads, etc. In addition, SDN offers a solution by linking ITS services to the vehicular network and reducing human intervention in the process of network management. The SDVN allows vehicular routing protocols to react to a dynamic environment and mobility of the vehicles. Furthermore, SDVN relieves the burden of open protocols to manage ITS applications. In fact, these open sources would run on top of the SDN controller through the northbound Application Programming Interface (APIs). Therefore, the features of SDVN would mitigate the issue of high mobility of vehicles, satisfy the requirements of ITS services, and enable heterogeneous radio access in the next generation vehicular networks.

## 3. The Architecture of Software-Defined Vehicular Cloud

In this section, we provide an extensive description of SDVC and its related functional layers. The SDN architecture of SDVC is presented. We further discuss the assistance of SDVC’s control plane to support potential cloud-based vehicular applications.

### 3.1. The Components of SDVC

The architecture for deploying cloud-based vehicular network based on software-defined networking, MEC cloud, and central cloud computing is described in [21,22]. Cloud computing and SDN are intended to work together by leveraging the use of a VM to improve connectivity and increase bandwidth. In fact, the concept of using SDN offers several imperative networking paradigms. SDN provides on-demand network programmable functions and adds meaningful scalability for deploying cloud-based applications on vehicles. SDVC supports the V2V communication through the concept of sharing the computing resources available on both cloud and MEC according to the resource management strategies. SDVC allows cloud vehicles to share three types of resource: computing, storage, and bandwidth resources.

Figure 5 shows SDVC architecture where vehicles are considered as SDN devices that are equipped with OpenFlow-enable software switches such as Open vSwitch [21]. The SDVC architecture in [21] supposes that SDN devices support virtualization technology and can host virtualization infrastructure (VI). VI will be able to install a cloud-based vehicular service on the vehicle. Here, a cloud-based application is considered as a service. When a vehicle requests the cloud-based vehicular service, the cloud instantiates a VM cluster and the MEC micro-data center located at the proximity of the vehicle loads the VM-Base. The VM cluster refers to a VI established in the central cloud [23]. The term MEC micro-data center means a batch of MEC servers that include virtualization infrastructure which provides computation, storage, and network resources for the cloud/edge-based vehicular applications. A dedicated VM (a customized VM-overlay) will be instantiated in the vehicle. The dedicated VM aims to provide computation resources of the cloud-based vehicular service. In the case the vehicle moves far away from the current MEC micro data center, the dedicated VM will be migrated to the nearest MEC micro-data center. This MEC micro-data center continuously offers that cloud-based vehicular service [4]. In addition, evolved Node Base (eNB) co-locates with RSUs, and other MEC equipment could host edge servers and radio access network computing in a cloud way. In fact, MEC computing is performed on a distributed topology [18] to deliver cloud-based vehicular services to vehicles. Certainly, MEC network resources could also be SDN devices or host SDN controllers. The VMs at the vehicle and MEC are named VM-overlay ( dedicated VM) and VM-base, respectively, as shown in Figure 3. A VM-overlay migrates and combines with the VM-Base hosted at other nearby MEC micro-data centers when it is time to move out of the communication range of the current MEC micro-data center. In addition, the VM cluster from the cloud resumes the VM-Base at the MEC micro-data center only when the vehicle cannot continue the trip due to an accident or any other barrier towards the destination direction.

Fault tolerance of one SDN controller requires a hierarchical design with SDN-enabled core network, SDN-enabled computation resources, and SDN-enabled access network [34,35]. Thus, in [21], the SDN controller is responsible for VM migration. The load balancing and routing decisions are abstracted through a central entity, referred to as the VC controller [22]. As shown on Figure 5, the SDN OpenFlow controller maintains a consistent view of the predicted trajectory to the destination. The SDN infrastructure communication controller preserves different VC established and allows VM services to resume at the next RSUs [22]. Allocation of virtualized resources pool and fault tolerance of a central controller influence the design of the SDVC with more than one central SDN controller. The active VM controller is responsible for managing the migration and replication of VM-overlay and VM-Base [21]. Here the management of VM includes replication, migration, and customization of a cloud-based vehicular service deployed as a dedicated VM.

#### 3.1.1. Multi-Access Edge Cloud in SDVC

The MEC computing micro-data center includes the control layer and the forwarding layer. The control layer is abstracted in the computing hardware, which includes the host operating system, a hypervisor, and other software entities based on VMs, namely active VM controller(s), SDN communication infrastructure controller(s), and SDN OpenFlow controller. The active VM controller controls the next replication and migration of VMs. The SDN communication infrastructure controller has the role to select the wireless access communication radio to connect the central cloud or MEC network resources during VM migration and resume procedures. The SDN OpenFlow Controller is hereafter considered as the main SDN controller. This SDN OpenFlow controller controls the global management of the SDVC network as well as the active VM and SDN communication infrastructure controllers. SDN OpenFlow controller enforces the management of the SDVC, as shown in Figure 6.

The forwarding layer includes forwarding devices such as an OpenFlow Wi-Fi access point and switch, local gateway, and local server service enablers. The VC supports V2X communications over C-V2X and Dedicated Short Range Communication (DSRC). The control plane decisions are not entirely made by a single SDN controller centralized element; adjacent SDN controllers in the MEC cloud could collaborate to make decisions. In fact, this collaborative decision is provided through mobility management, and Quality of Service (QoS) support provided at the SDN OpenFlow controller. The SDN OpenFlow controller should achieve a fast recovery after failures between adjacent controllers in the SDVC. A logically distributed control plane assisted by mobility management achieves optimum performance and robustness against failures [36].

#### 3.1.2. V2X Cloud in SDVC

The V2X access cloud includes vehicles and road users. Moreover, SDVC includes infrastructures to infrastructure links, a vehicle to network links, a vehicle to infrastructure links, a vehicle to vehicle links, vehicle to network links, and vehicle to vulnerable road users links. In addition, the V2X access cloud includes vehicles and UE-type RSU, which are considered as SDN devices similar to data planes in the SDN concept. Functions of the SDN devices in the V2X cloud are concentrated on data collection, forwarding, queuing, and sending data into the MEC micro-data center. Therefore, the vehicle-enabled V2X could be built with the following functional modules:Information collection module of vehicles: The information collection module includes different types of sensors (safety and diagnostics sensors) in the vehicles. Availability of these sensors in the vehicle allows the information such as speed and revolutions per minute (RPM). Furthermore, surrounding or adjacent vehicles, pedestrians on the roads information, and weather conditions are collected for SDVC.V2X application module of vehicle: This module is considered to act as the traditional On-Board Unit (OBU) integrated in-vehicle wireless communication interface. It allows broadcasting information such as geo-messaging to MEC micro-data center to notify when the vehicles enter or exit certain regions of the roads. The V2X application of the vehicle will receive the message corresponding to the routing rules defined by the SDVC’s SDN OpenFlow controller. Moreover, the V2X application module receives and stores the current position information of the vehicle provided through the global position system (GPS) [37].Communication capable 3GPP LTE UE module of vehicle: The communication module includes the V2I, V2V, and V2N communications modules. V2I, V2V, and V2N communications should comply with both the requirements of LTE and service requirements for cloud-based V2X services. The V2V communication requires the communication module of a vehicle to get an exact subscription and authorization from a network operator. The communication module of the vehicle allows V2V applications to transmit no safety messages such as locations. The V2V communication module provides direct communication (LTE PC5) as it is defined under Release 12’s proximity services communications (“ProSe”) function [38]. Radio communication interference caused by direct interoperability communication between the vehicle’s communication interface limits the transport of information. The vehicle should use either cellular infrastructure (LTE-Uu) supporting V2X communication or DSRC depending on C-ITS safety messages.

Enhanced node BS (eNBs) type RSU deployment provides cellular communications to vehicles. In SDVC, UE-type RSU transmits wireless signals from or to the communication module of the vehicle through LTE-Uu. The communication module of UE-type RSU sends unicast messages to a MEC computing server. Furthermore, the information provided by the central cloud are distributed to all the vehicles into the relevant geographical area through either UE-type RSU or evolved Multimedia Broadcast Multicast Service (eMBMS).

#### 3.1.3. Components of the Multi-Access Edge Cloud-Enabled SDN for VM Migration

Figure 6 shows the components of the multi-access edge (MEC) cloud assisted by SDN for supporting VM migration over SDVC. The SDN concept envisions an architecture of three layers: data plane, control plane, and application plane. In SDVC, the MEC cloud is assisted by a control layer that includes the SDN controllers and applications on top of the control plane to allow programmable functions to assist the SDN controller to greatly manage the SDVC network. The data plane includes the forwarding devices considered as SDN devices. The vehicle has physical hardware that hosts VMs through hypervisor software. The OBU equipped in the vehicle has wireless interface for communication. The OBU in vehicle communicates with the MEC infrastructure when the routing rules are configured in their SDN device’s flow tables; otherwise, the unmatched packets are directly forwarded to the SDN controller or dropped according the actions field in SDN devices’ flow tables. As the vehicle moves out of the radio coverage offered by the MEC nodes, the latter should be able to keep data and running services on the OBU through VM migration. The VM migration and routing paths among MEC nodes are managed and controlled by the control plane in SDVC. In SDVC, vehicle mobility and heterogeneity of wireless technologies equipped in vehicles introduce further challenges during VM migration process. The updated information about the topology changes, vehicle mobility, network interfaces, and information on different active VM (ready to be migrated) are primordial to the control layer since the coordination and management are subjected to the SDN controller and applications involved in supporting the SDN controller. To efficiently migrate VM from one MEC micro-data center, the SDN control has a communication infrastructure controller to manage the topology and network preferences wireless interface equipped with vehicle node that hosts the dedicated VM. The latter migrates and resumes its computation to the next adjacent MEC micro-data center. On the other hand, the active VM controller performs a significant role regarding control of the VM migration process of VM services on vehicles. Based on the data collected from the SDN communication infrastructure controller, the VM migration of active VM controller is decided. The decision is about whether an active VM controller should be migrated or not along with the vehicle. Therefore, the SDN controller should minimize the overall network cost of VM migration. As shown in Figure 6, the control plane includes the MEC server. The MEC servers implements the features adopted by the ETSI [39]. The MEC server hosts applications designed to provide latency and high bandwidth as well as real-time access to radio network information. In the context of VM migration assisted by the SDN controller, imperatives applications modules such as V2X collaborative sharing and SDN overlay VM migration management are responsible for resources sharing and VM migration, respectively. These two applications would be added among other validated applications of the MEC application platform. In the application plane, existing applications such as quality of services, mobility management, load balancing, and content-based routing are designed to provide applications programming interfaces to manage efficiency quality of services, mobility, load balancing, and routing based on the data content requested by the vehicular user. The application plane interacts with the control plane through northbound API, while the southbound API implements control policies between the control plane and the data plane. The OpenFlow protocol maintains a secure communication between the control plane and data plane.

In general, SDN control in SDVC has the responsibility to harmonize the centralization concept of the SDN control with the native VC. A hierarchical control solution with a communication infrastructure controller and the active VM controller can be considered. The optimal placement of the controllers, routing overhead between them (active VM, communication infrastructure, and SDN OpenFlow controllers) challenge the management of VM migration service. In fact, the inter-controller communication throughput between the adjacent SDN controllers would be maximized through optimum algorithm pertaining to the main SDN controller’s placement with active VM and SDN communication infrastructure controllers.

The SDN OpenFlow controller controls and forwards routing rules for, e.g., deploying any ITS services initiated from cloud computing. These rules are pushed to the SDN communication and infrastructure controllers through the northbound interface. The core architecture of the SDVC is adopted to logically centralize data forwarding and computation resources. The ETSI framework defines a reference for their deployment of V2X services that require ultra-low latency access [39]. The access of the V2X application is done through the MEC server. MEC servers provide mobile edge (ME) computing resources such as computation, storage, and network. Furthermore, MEC is an entity that contains the ME platform.

### 3.2. SDN Controller Assistance for Cloud-Based Vehicular Applications

Cloud-based vehicular applications rely on resource computation, storage, and bandwidth sharing to deliver services in terms of VM to the vehicles requesting cloud-based vehicular applications. A discussion of some cloud vehicular applications over the SDVC is given in the subsections below.

#### 3.2.1. Cloud Vehicular Application Based on Computation Resource Sharing

Real-time navigation with computation resource sharing serves as a hypothetical use case to illustrate the potential benefits of applications of cloud-based vehicular networks [4]. A vehicle is using real-time navigation controlled from the central cloud. It will first request cloud service from the central cloud and MEC cloud. The SDN OpenFlow controller controls virtualization infrastructure of MEC micro-data center towards the driving direction. SDN OpenFlow controller also controls the VM cluster from the cloud to provide the navigation cloud service by suggesting several routes based on the current traffic conditions. The SDN communication infrastructure controller acts as an agent to push control traffic to the vehicle considered as a SDN device. The active VM controller updates the vehicle with navigation paths conditions on the road and controls the migration of dedicated VM to the next MEC micro-data center. After the vehicle moves out of the radio range of the current serving MEC, the vehicle resumes dedicated VM independently of the radio access at the next MEC micro-data center. During the trip, traffic congestion or an accident could happen, which requires the SDN OpenFlow controller to suggest routing decisions. Consequently, the VM cluster in the central cloud updates new navigation directions before vehicle entering the congestion area.

#### 3.2.2. Cloud Vehicular Application Based on Storage Resource Sharing

Storage resource sharing is an application where many data are stored on independent MEC storage resources. For example, video surveillance on a public transport bus uses high definition cameras to record real-time surveillance inside the public transport bus. This means the requirement of installing higher storage units. However, the cost of higher capacity units is not a suitable option. Cooperative storage is a valuable solution. The public transport bus accesses multiple MEC clouds toward its direction and stores the recorded real-time videos to its nearby MEC micro-data center’s server. The recorded videos are deleted either when the public transit bus finishes uploading them or when it drives towards the next MEC micro-data center. The SDN communication infrastructure controller provides the location of the next MEC micro-data center to record (store) recorded videos. SDN communication infrastructure controller determines the next location based on a set of information such as speed limit, available size of storage resource at the MEC micro-data center, and the access radio of the MEC micro-data center. With this set of information, the active VM controller would migrate the storage as service only to a MEC micro-data center that has enough storage space. The SDN OpenFlow controller updates routing path of MEC micro-data center that has in-vehicle video surveillance files. Content-based routing modules at the application layer would suggest the SDN OpenFlow controller to define routing rules so that the road operators would easily permanently retrieve at a given time a different slice of video stored on different MEC micro-data center. At the time of an accident, the road authorities would request the active VM controller to initiate routing rules to allow the transportation bus to upload the recent video not yet stored on the MEC micro-data center.

#### 3.2.3. Cloud Vehicular Application Based on Bandwidth Resource Sharing

Downloading from a central cloud a document by a passenger in vehicle is bandwidth consuming. In cloud vehicular network, a cooperative downloading is an option for allowing vehicles driving in the same direction to help other passengers download a large file. Vehicles in a group of V2X cloud download the file. For this scenario, a distributed active VM controller is established to control the cluster of vehicles downloading the file from the cloud. The SDN communication infrastructure controller relies on the vehicle collaboration sharing function (function at application plane) to update flow tables of the OpenFlow switch directly connected to vehicles with their vehicle Identification (ID). The vehicle collaboration sharing function enables bandwidth sharing application. In fact, this service accelerates the rate of network bandwidth by speeding up the download compared to one vehicle handling the download task. Thus, the active VM controller assists each vehicle (involved in downloading the file) to transmit its own part to the owner, which in turns merges (reassembles) all other downloaded parts to get the entire file.

The management of cloud-based vehicular applications based on storage, computation, and bandwidth sharing are still worth discussing in the next generation vehicular network. Emergent technologies such as SDN and Network Function Virtualization (NFV) have potential to address the issue of ultra-low latency for cloud-based vehicular applications. Certainly, SDN and NFV would influence the deployment of cloud-based vehicular applications for the next generation vehicular network. Table 1 summarizes potential applications in cloud-based vehicular networks supporting SDVC. We also provide relevant SDN controllers on the control layer to assist each application. Zhang et al. [4] first discussed the resource sharing assistance.

## 4. Migration Management of VM over SDVC

This section provides the assistance of migration management of VM through the SDVC’s control plane. Four cases of vehicle mobility in VC are described. We further provide a review of existing frameworks of SD-based vehicular computing to assist VM migration approach. In addition, we discuss the state-of-the art of existing VM migration techniques. Furthermore, we present a comparison review that focuses mainly on the network emulators/simulators to evaluate SDVC’s use cases.

### 4.1. SDN Controller Assistance for VM Migration over SDVC

The MEC cloud consists of the control layer and the forwarding layer. The assistance for VM migration is controlled through the control layer, which includes several SDN controllers. The SDN controller holds the management of the global network. It requires getting network status from SDN communication and active VM controller. The SDN communication infrastructure controller is in charge of adjacent MEC micro-data center and particularly manage the connectivity of heterogeneous radio access when the VM service resumes the adjacent MEC micro-data center. The migration of VM-overlay and VM-base is managed by the active VM controller as long as the vehicle moves from the current MEC micro-data center hosting the cloud-based vehicular services. The active VM controller manages the flow tables of MEC radio access and forwarding devices at the forwarding layer.

The MEC micro-data center supports closest vehicles (V2X enabled UE). As the coverage of MEC micro-data center would experience short disconnection in case the vehicle moves far of the MEC radio access coverage. In [4], four scenarios of VM service migration on cloud-based vehicular network have been proposed. Based on the four VM migration case studies (Inter-Cloudlet, Intra-Cloudlet, Across Roadside-Vehicular Cloud and Across Roadside-Central Cloud) of Zhang et al. [4] and the concept that routing policies of RSU and vehicles are managed from the control plane, this paper discusses the VM migration of active VM controller.

The inter-MEC migration service concerns the VM migration from one MEC micro-data center to its immediate next MEC micro-data center when the two MEC computing resources are in different radio access network coverage. In this case, the active VM controller of MEC micro-data center-1 will not be migrated because the dedicated VM migrates to the next MEC micro-data center. Therefore, the active VM controller of MEC micro-datacenter would include routing paths to new packets from central cloud service. Then, the forwarding device of the radio access would take actions regarding the type of VM migration pre-defined in the in-coming packets field of flow entries. This scenario is shown on Figure 7a.

Figure 7b shows the intra-MEC VM migration case. The adjacent MEC micro-data enter connects to the same radio access network. In this case, the active VM controller would migrate at the same time with VM-operating system/application. The active VM controller should continue to manage forwarding devices in the two MEC micro-data center since they share the same radio access network.

In the case of across MEC micro-data center computing, vehicles establish an ad-hoc network. In this case, the access radio coverage is the same for the two MEC micro-data center. During the trip, vehicle A accelerates at the point to break the ad-hoc connection. Vehicle C will access the radio access because vehicle D still hosts a VC with enough computing and network resources. The active VM migration on micro-data center-2 will then update routing rules to SDN devices on vehicle D so that vehicle C would continue hosting the dedicated VM of the current cloud-based vehicular service. In this scenario, the active VM in micro-data center-1 would continue control the routing rules of vehicle A. The SDN communication infrastructure controller would assume that vehicles D and C connect to the access radio deployed at MEC micro-data center-2. The SDN communication infrastructure controller in proximity would then allow the active VM controller-2 to establish forwarding rules to the rest of vehicles in the ad-hoc network established. This VM migration scenario is shown in Figure 7c.

The other case is shown in Figure 7d and concerns across MEC and central cloud. This case is similar to the previous but that, after a while, most of the vehicles leave the existing ad-hoc network. Therefore, there is, for instance, no radio access network to connect vehicle C. In fact, vehicle C establishes the connection with the central cloud. This means that the dedicated VM is migrated at the central cloud. In this case, the SDN OpenFlow controller updates routing rules to both SDN communications infrastructure controller and the active VM controller according to the location of vehicle C.

### 4.2. Review of the State-of-the-Art of VM Migration Techniques over SDVC

#### 4.2.1. Overview of Architectures and Technologies Integrated with SDN and VM Migration Approach in SDVC

SDVC architectures integrate fog/edge computing to keep data and processing services as close as possible to the vehicular users but slight VM migration strategies, as presented in Table 2. In [9], a latency performance evaluation was discussed for processing latency sensitive services using the construction model of the software-defined cloud/fog network graph model. Furthermore, edge computing approach for delivering traffic content query requested by vehicles was proposed in [45]. The approach in [45] enabled vehicles to communicate with neighboring ones through vehicle-level caching technique to minimize the network latency. Automation of traffic, improvement on disseminating driver and pedestrian safety messages, and monitoring of traffic violations are implemented through an intelligent transport system based on MEC and SDN technologies [46]. The architecture improves routing data. Furthermore, offloading algorithms achieve better packet delivery time and packet loss. The authors of [47] proposed the conceptual principles that integrate the fog/cloud network to improve user experiences and reliable vehicular communications in order to support vehicular applications with restricting QoS/QoE requirements.

Cooperative driving services are enabled by architecture framework based on SDN, NFV, and MEC to support V2X network slices. In [53], distributed pseudonym pools are managed and scheduled through software-defined pseudonym system. The SDN concept mitigated the system overheads cost by proposing two-side matching theory that solves the matching problem among the pseudonym pools. In the works by Campolo et al. [52], V2X network slices are implemented through the use of softwarization technologies such as SDN, NFV, and MEC. The aim of the V2X network slice enables the cooperative driving services under intra- and inter-operator mobility. The problem of mobility is addressed by the use of V2X network slices. It is realistic to improve the packet delivery rate of safety messages through VM migration services. In addition, the SDN controller selects the best MEC on which a current cooperative service would resume to the next MEC server. Autonomous vehicular networks provide significant cooperative services for Connected and Autonomous Vehicles (CAVs) [51]. The regular continuation of computing and storing resources services at the edge of the core network enable NFV and SDN to provide different applications with provisioning cost. Softwarization at the edge of the network improves the intelligent traffic steering, management of multiple computing resources, and reliable access to the network through different wireless access technologies.

Service migration strategies based on SDN technology enhance delivery of cloud-based services to mobile users and vehicular users as well. Tarik et al. addresses user mobility and its related challenges by designing a migration strategy called Follow-Me Cloud [49]. The Follow-Me Cloud prototype offers an optimized user experience because the migration of current service is triggered by the Follow-Me controller. The latter uses the user current location and movement direction. During the user movements, the ongoing service hosted on the current MEC server is migrated once an optimal decisions predicted by the Follow-Me controller is delivered to the mobile operators. The software-defined networking and locator–identifier separation protocol-based improve mobile cloud-based service. The Follow-Me concept introduces optimized migration models which might be implemented as applications modules to support the SDN controller to select algorithm for service migration decisions. The Follow-Me concept handles the tradeoff between migration cost and user experience. Reconfiguration overhead related to VM migration was proposed by Salahuddin et al. The authors proposed a RSU cloud composed of traditional RSUs and micro-data centers [50]. The RSU cloud is designed to use SDN and can host RSU services to handle drivers requests. During congestion time that leads to inherent dynamic demands from the vehicles, RSU cloud is reconfigured to optimally meet the service demands. The optimization is based on the reinforcement learning algorithm to select a reconfiguration overhead that minimizes the VM migration cost.

#### 4.2.2. Simulation Tools for VM Migration Services over SDVC

The SD-based vehicular cloud still lacks real simulation environment tools to better evaluate the performance of management of VM migration. The Mininet-WiFi [54] lacks cloud-based functions to ensure the simulation of VM migration. The work in [36] proposes a base foundation for the design and implementation of an SDVN-based cloud simulator. In addition, mobility in the vehicular network affects the migration of VM. Therefore, SDVN-based cloud simulator and existing traffic simulator such as Simulation for Urban Mobility (SUMO) [55] would support the evaluation of VM strategies in SDVC. The SDN-based cloud simulator integrated with traffic mobility of vehicle nodes is addressed as research directions for future works. Most SDVN enabled edge/cloud computation methods address the issue of resource management to ensure higher network performance. Multi-access edge computing would address the issue of management of VM Migration (VMM) in order to allow VM to resume themselves.

Simulator tools for the cloud network model based SDN controller are proposed in [56,57]. These simulator tools only support simulations scenarios for fixed cloud SDN-controller networks and usually run single- or multi-VM hosts. A survey on simulation evaluation of network model software-defined vehicular based edge (fog) and cloud computing network model is given in Table 2. The migration of VM is performed for keeping the computation service deployed at either edge or cloud computing infrastructure. Mininet-WiFi [54] and OMNeT++ [58] integrated with Simulation for Urban Mobility (SUMO) [55] are the most currently used to evaluate the performance of SD-based vehicular cloud system. Mininet-WiFi is an extension of the Mininet [59] emulator to simulate a mobile wireless node. SUMO is used to inject traffic mobility to allow Mininet-WiFi to evaluate SDVN architecture [45,52]. However, the Mininet-WiFi platform lacks the ability to apply VM migration technologies. A third platform to evaluate VM migration was developed by Atwal et al. [36]. The platform in [36] evaluates cloud-based applications but does not propose a new approach for the migration of VM. It is clear that a platform for cloud and virtualization concepts integrated to Mininet-WiFi is needed. Part of Mininet-WiFi, the OMNeT++ network, evaluates a SDVN-based edge computing in [47]. OMNeT++ [58] has modules to emulate cloud computation particularly for realistic VM migration simulations environments. Accordingly, the emulator platform that evaluates smart cloud applications [36] should be extended with a traffic mobility interface to improve the simulation evaluation of SD-based vehicular cloud applications.

The lack of an SDN-based vehicular cloud simulator limits the performance of the control plane in the SDVN system. Table 2 shows that most SD-based vehicular and edge cloud computing concentrates on mathematical analysis rather than to realistic simulation environments. A vehicular edge/fog computing to support the control layer over SDVN for content delivery is discussed in [9,45,46,47,48]. Al-Badarneh et al. [45] and Campolo et al. [52] evaluated slicing road framework in 5G network softwarization for content delivery using Mininet-WiFi emulator. The content delivery is transmitted through V2X communication. Furthermore, He et al. [9], Nobre et al. [47], and Borcoci et al. [48] used mathematical analysis to compare the proposed scheme for content delivery. The lack of a realistic simulation tools that support OpenFlow-based edge computing coupled with vehicle traffic mobility affect the evaluation of VM migration techniques in SDVC.

#### 4.2.3. Strategies for Virtual Machine and Service Migration in Vehicular Cloud and Edge/fog Computing

Table 3 recapitulates some strategies for VM service migration in vehicle cloud and fog computing-based architecture for cloud-based applications.

Fog computing addresses the issue of computation latency by leveraging service of VM migration assisted with migration policies. Strategies for service migration such as Follow-Me Edge, pre-copy based Live VM, and location-based VMM are implemented in [61,62,69], respectively. Follow-Me Edge presents a concept that enables services to move across edge servers as per the movement directions of their respective users. This concept mitigates VM migration latency by leveraging migrations policies implemented at the edge servers. Furthermore, pre-copy based Live VM is designed to reduce both migration time and downtime. This approach relies on dynamic approach that executes a regression analysis based on the amount of dirty pages recorded previously to predict the downtime. The prediction of the downtime value is compared with a predefined downtime to determine the right stage to execute a movement into the stop and copy stage. The location-based VMM considers the fog computing paradigm and its features to keep processing of cloud based applications in the proximity of users. This strategy identifies business models to keep VM migration reliable, particularly as the mobile users move among fog servers. Some of the proposed business models are: Service Provider/Service Orientation, Support and service contracts, and All-in-one enterprise cloud [69].

Strategies for service migration in no vehicular environments depend on mobility prediction solutions to enhance fog computing resources and global network traffic. MyiFogSim evaluates resources allocation in fog computing assisted by a user mobility prediction strategy while a VM service is migrating among fog servers towards users’s directions [60]. MyiFogSim simulator defines migration policies that examine when a user should be migrated and where the user’s VM is going, i.e. the destination of fog and how the migration is handled. In [63], a proactive replication mechanism is introduced as a promising VM strategy to avoid QoE degradation during service migration between different edge nodes. The proactive replication is assisted by two linear optimization algorithms for replication-based service migration. Markov Decision Process (MDP)-based service migration is introduced in [65]. MDP is the analytical model used to evaluate the performance of the Follow-Me Controller (FMC) [49]. Hence, MDP model mainly focuses on finding the user equipment (UE) position regarding the optimal data center (DC). The time window based service migration aims to find the optimal service placement so that the average cost of VM migration over a given time would be minimized [66].

Vehicular cloud computing uses mobility prediction not only to allow service migration of VM running RSU/cloud services but also mitigate the challenge of high mobility and heterogeneity of wireless interface equipped in vehicles. Yao et al. [64] introduced the Road Side Cloudlet (RSC) in RSU cloud established between the vehicular cloud and central cloud computing. RSC-based VM service relieved the challenge of the low response-time problem and the high communication cost experienced for vehicular users requesting cloud-based vehicular applications. The authors of [64] considered that RSC-based VM service has a very high execution cost in long-distance networks between vehicular users and RSC. Therefore, two heuristic algorithms are proposed in [64] to improve VM migration cost among RSCs. The first relies on an algorithm that finds the shortest path according to the vehicle movement towards to the next RSC without dropping link robustness. The second heuristic algorithm is based on VM allocation cost in RSC. Vehicular Virtual Machine Migration (VMMM) was proposed by Refaat et al. [68] to address the issue of vehicle mobility for vehicles considered as VM hosts in the vehicular cloud. The authors of [68] introduced factors that affect the vehicle to successfully complete the migration of VM. These factors are: geographical mapping of the wireless coverage, congestion of the vehicles, vehicle speeds, and wireless communications technology equipped in vehicles. Based on these four factors, the authors proposed three migrations schemes: (1) VVMM-U (uniform) that includes vehicles migrating after the proposed VVMM algorithm selects the optimal vehicle destination; (2) VVMM-LW (Least Workload) that includes a vehicle that switches its workload by selecting the next destination to resume the workload using the proposed VVMM algorithm; and (3) VVMM-MA (Mobility Aware) that includes vehicles that entrust the mobility awareness of neighboring vehicles before attempting to migrate.

## 5. Conclusions

In this paper, we summarize our findings along with lessons learned collected from our study on the state-of-the art accomplishment of SDVC-related cloud-vehicular applications and their VM management with other technologies (e.g., fog computing, Follow-Me cloud/edge, and vehicular cloud) for offering cloud-based vehicular applications. Based on the many substantial research works we reviewed, academia and industry are proposing design and configurations of SDVC architectures that would support cloud-vehicular applications. Therefore, efficient resource management, low cost VM migrations and processing latency, are stringent requirements for delivering cloud-based applications. Although various VM migrations strategies have been proposed in vehicular cloud computing, the extensive investigation to evaluate the effectiveness of VM migration management of the SDVC remains an open issue. The researchers envision the use of SDVCs coupled with other technologies such as edge/fog computing and NFV. The control layer of SDVC would play an important role in the management of computing resources. In addition, research directions should focus on the implementation of intelligent selection algorithms to run inside the control layer of the SDN controllers in SDVC. For VM machine migration, when the vehicle decides to migrate the service, the MEC may require the SDN controller to choose the most suitable MEC server to start the service migration process. The SDN controller should be able to estimate the potential cost incurred and compare it against the resource utilization modules designed to improve the MEC cost and QoS from the point of view of end users. VM migration process in SDVC architecture can be further modeled using existing VM strategies discussed and implemented as applications that consider QoE requirements. The selection algorithms based on SDN concept would help to produce the best routing policy to decide whether to migrate a service or not. QoS-aware mobility multi-access selection algorithm; selection algorithm of migration path with both cost; resource allocation on multi-access edge servers; development of a high service migration mechanism for inter-MEC migration case, intra-MEC VM migration case, and micro-data computing across MEC; and central cloud to ensure service continuity are considered as research directions. In addition, an emulation platform for evaluating QoS and routing applications of SDVC applications is missing. The real traffic mobility using SUMO, for instance, might be included to evaluate the simulation of SD-based vehicular cloud applications. To support a range of unprecedented cloud-based vehicular applications that improve connectivity and bandwidth of cloud-based applications, SDVC is introduced. SDVC architecture considers vehicles to host virtualization services for enabling a VM to migrate from one RSU to another. In addition, we studied the evolution of computation and networking technologies of SDVC with a focus on its architecture within the cloud-based vehicular environment. Then, we discussed the potential cloud-based vehicular applications assisted by the SDVC along with its ability to manage multiple VM migrations. We also provided a detailed comparison of existing frameworks in SDVC that integrate the VM migration approach and different emulator or simulator networks used to evaluate VM frameworks’ use cases. In future work, we aim to provide further evaluation of discussed schemes based on their theoretical concepts and their performance. We will also consider the network resource slicing at the multi-access edge computing cloud for densely deployed vehicular network scenarios.

## Figures and Tables

**Figure 1 sensors-20-01092-f001:**
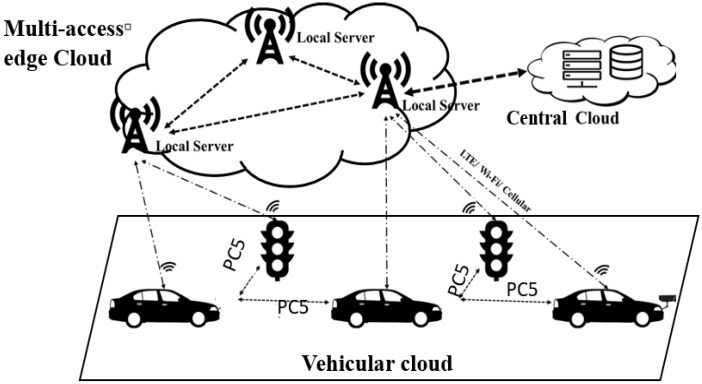
High level of cloud-based vehicular application network.

**Figure 2 sensors-20-01092-f002:**
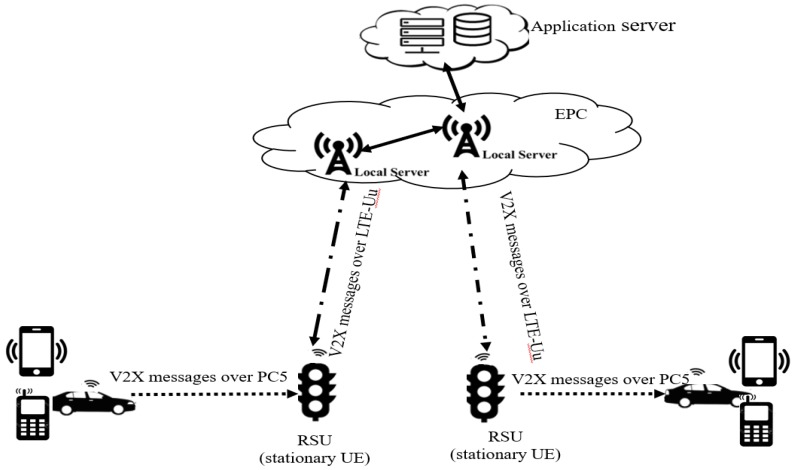
V2X services based on two working modes of cellular network.

**Figure 3 sensors-20-01092-f003:**
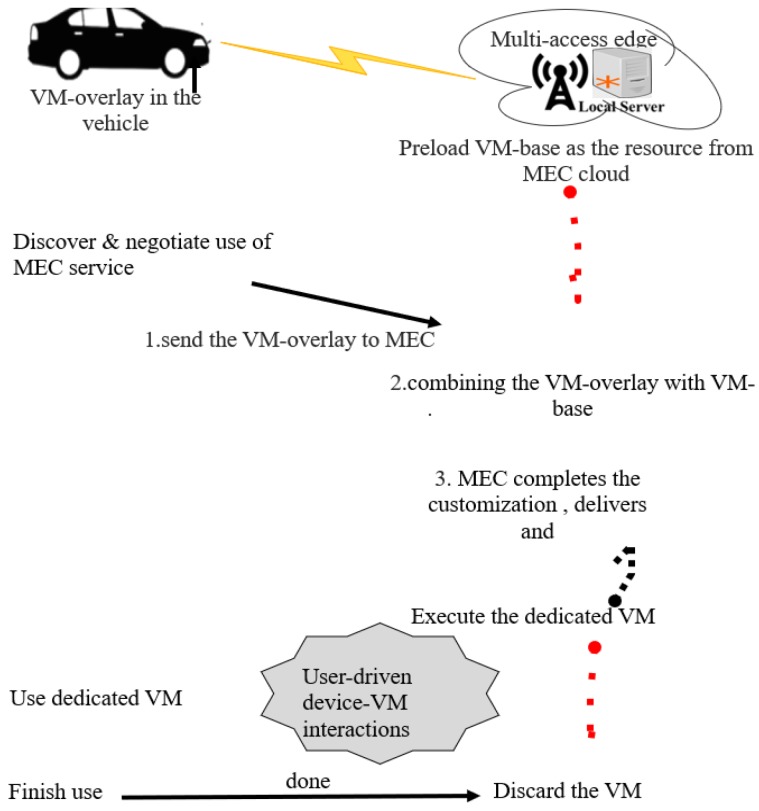
Dynamic VM synthesis approach to instantiate a VM for a cloud service at MEC cloud.

**Figure 4 sensors-20-01092-f004:**
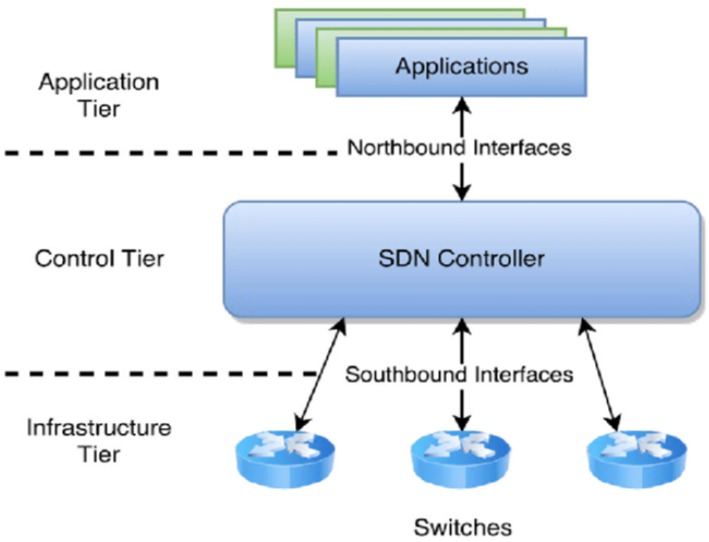
Logical View of Software-defined Network.

**Figure 5 sensors-20-01092-f005:**
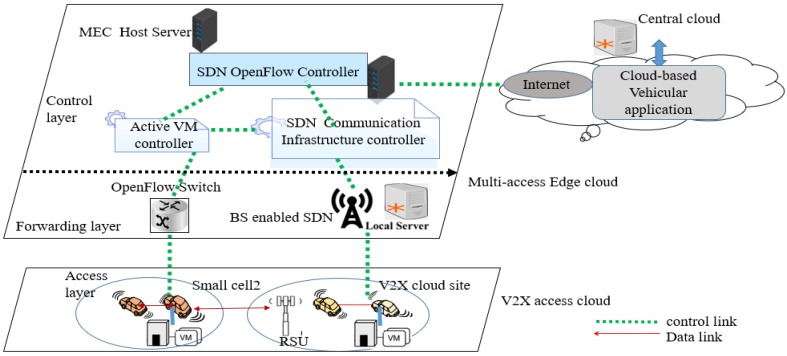
Software-defined vehicular cloud network architecture.

**Figure 6 sensors-20-01092-f006:**
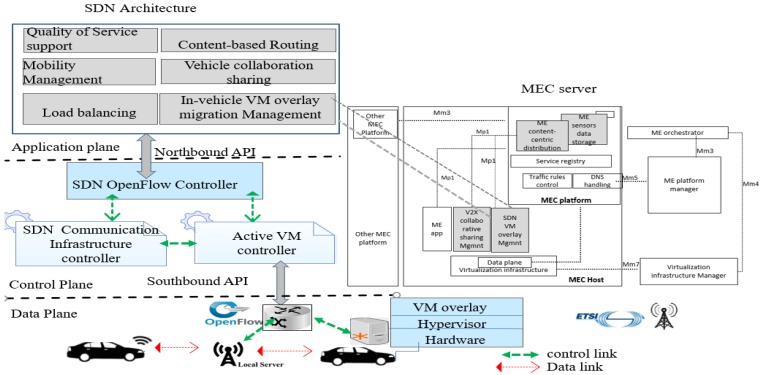
Multi-access edge cloud-enabled SDN (left) and MEC server (right) for VM migration.

**Figure 7 sensors-20-01092-f007:**
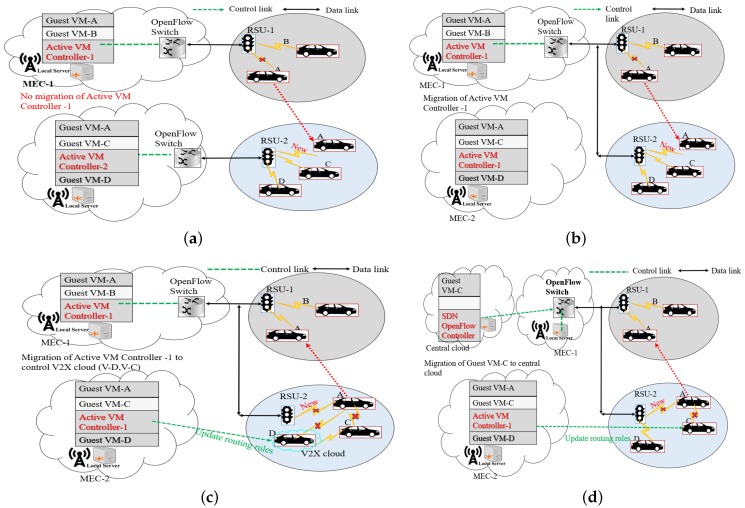
Active Controller migration scenario in the Software cloud vehicular network: (**a**) inter-MEC migration case; (**b**) intra-MEC VM migration case; (**c**) across MEC micro-data computing; and (**d**) across MEC and central cloud.

**Table 1 sensors-20-01092-t001:** Applications based vehicular network over SDVC.

	Relevant SDN Controllers Assistance			Resource Sharing			
Potential Application	SDN OpenFlow Controller	SDN Communication Infrastructure Controller	Active VM Controller	Computation	Storage	Bandwidth	Latency (ms)
Real-time traffic condition analysis and broadcast	√	√	√	√	√		10–100 [40]
Real-time car navigation	√	√	√	√			430–460 [41]
Video surveillance		√	√		√		120–200 [42]
LBS commercial advertisement		√	√		√	√	10–100 [40]
Mobile Social networking	√	√	√	√		√	10–100 [43]
In-vehicle multimedia entertainment		√	√	√	√	√	10–100 [40]
Intervehicle video and audio communications			√			√	10–100 [43]
Remote Vehicle Diagnostics	√	√	√	√			10–100 [40]
location-based services recommendation based on personalized and precise service [44]	√	√	√	√	√		10–65 [44]

**Table 2 sensors-20-01092-t002:** Review of the state-of-the-art of VM migration techniques over SDVC.

SDVN Proposal and Its Computing Resources Architecture	Year	Approach to VM Migration	Simulation Tools	Use Cases
[21], central cloud	2017	none	not specified	update softwares on vehicles
[48], Fog computing	2017	none	not specified	geographic placement of the SDN controllers
[36], central cloud	2018	none	Mininet-Wifi and emulation platform for evaluation of smart cloud applications	QoS and routing applications
[49], central cloud	2016	Fellow Me Cloud prototype	software-defined networking-based follow-me cloud testbed	evolution of service latency during and after the migration process
[50], RSU cloud	2015	reconfiguration overhead related to VM migration	Mininet	not specified
[51], multi-access edge computing	2019	none	not specified	bandwidth management for Connected Autonomous Vehicle
[52], multi-access edge computing	2018	none	Minenet-WiFi	V2X slice for cooperative driving services
[53], vehicular cloud	2016	none	none	not specified
[47], fog computing	2018	none	OMNeT++, SUMO [55]	content distribution
[46], edge computing	2018	none	not specified	content dissemination
[45], edge computing	2018	none	Mininet-WiFi, SUMO	content distribution
[9], edge computing	2016	none	not specified	content dissemination

**Table 3 sensors-20-01092-t003:** Virtual Machine migration techniques in Vehicle cloud computing.

Strategies for VMM	Mobility Prediction	Vehicular Cloud	Cost on VMM	RSU	Fog Computing	Migration Policies	SDN Concept	Reduce VM Failures	Resources Management
MyiFogSim [60]	√		√		√			√	√
Fellow-Me Edge [61]			√		√	√		√	√
pre-copy based Live VM [62]					√			√	√
proactive service replication [63]	√				√	√		√	√
VM-based RSUCloudlet [4,64]		√	√	√				√	
MDP-based service Migration [49,65]	√		√		√	√			√
Time window based Service Migration [66]	√		√		√			√	√
Predictive VMM [67]	√	√		√		√		√	√
Vehicular Virtual Machine Migration [68]	√	√	√			√		√	√
location-based VMM [69]			√		√	√			√

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
