# Peer review of "Software-Defined Vehicular Cloud Networks: Architecture, Applications and Virtual Machine Migration"

_sensors, 2020, doi:10.3390/s20041092_

Round 1

Reviewer 1 Report

Authors present an extensive discussion of the state of the art for vehicular cloud networks. There are parts of the work that convey a thorough and relevant discussion which can significantly improve the paper if these were shifted towards the beginning. For example, Section 3.2 discusses the main problem addressed in the paper: the role of migration in cloud resource sharing for vehicular networks. Although the four figures discussed (within that section) are pertinent and summarize well the different scenarios for VM migration, the figure lacks clarity, proper font sizes and readability. This figure does not connect to every aspect to the architecture discussed at the beginning of the paper, ie, the one discussed in Fig 1. So it should be improved. Moreover, figures 1, 2 and 3 should be homogenised in every aspect, ie, using the same icons and the same system components, etc. (also in the discourse). Please pay special attention to Fig 3, the three different steps are confusing and are not properly discussed in section 2.3. Also, the last paragraph of that same section does not fit - I'd suggest eliminating it altogether. 

There are other aspects that should be considered throughout the paper:

repetition of references in the same citation  fallacies in statements: "This is true because some papers say so..." Fig 6 is overly complicated and does not convey a clear message, neither is it discussed in the same extent (as the complexity) within Section 3.  You can separate in different subsections the section 3.1, one starting at line 303, another one starting at line 322, and the last one starting at line 352. Please consider adding a column with latency boundaries to Table 1 to convey a more pertinent and precise message regarding the type of applications.  Although you cite an extensive number of papers in section 4.2, you let the reader believe that you are about to discuss a simulation model when you only present the state of the art to argue that the research community lacks experimental results coming from trustable implementations/models. I'd advise changing the tone of the discussion. Conclusions must be rewritten. You should summarise what the research directions are, what the weak and strong points of current research are, where the emphasis of research should be from your perspective, etc.  

Author Response

Find here attached  cover letter to explain point-by-point the
details of the revisions in the manuscript and our responses to the
reviewer 1' comments.

Reviewer 2 Report

For vehicular cloud networks, this work attempts to propose SDN-based architecture, discuss the applications and investigate the VM migration method. However, there are some suggestions for improving this work. 1. There are too many grammar mistakes. For example, “due” must be “due to”, “the cloud computing help to…” must be “the cloud computing helps to…”, “such” must be “such as”. 2. The texts in Figures look blurry, especially, in Figure 7. The readers cannot see them clearly in printed version without colors. The authors should re-draw the figures in easily distinguished styles. 3. The latest research achievements must be selected for comparison studies in the manner of quantitative numerical analysis, especially, the VM migration method. 4. The authors should better and more clearly point out which are the exact elements of technical novelty of their solution. The current presentation seems to be a pure description of engineering, but is lack of theoretical depth. 5. The reviews may be insufficient. Few important papers are missing in the literature which could be added to introduce the applications of IoVs. * Ming Tao, Wenhong Wei, Shuqiang Huang. Location-based Trustworthy Services Recommendation in Cooperative-Communication-Enabled Internet of Vehicles. Journal of Network and Computer Applications, Volume 126, 15 January 2019, Pages 1-11.

Author Response

Find here attached  cover letter to explain point-by-point the
details of the revisions in the manuscript and our responses to the
reviewer 2' comments.

Reviewer 3 Report

    In order to improve the connectivity and bandwidth of cloud-based vehicular applications, a software-defined vehicular cloud network was introduced. The new architecture and potential cloud-based vehicle applications were studied and discussed. The management scheme and theory of virtual machine in the new architecture were put forward. Overall, I found the work was interesting and of great theoretical significance. However, there are still some imperfections in the work. I hope that my comments would be useful for improving the quality of the paper. Some of the detailed comments are as follows:

In order to make the work easier to follow, two suggestions may play a certain part:

1) Fully and carefully re-edited by a native English-speaking specialist is needed;

2) The chart representation should be further increased to enrich the expression form of the article.

More theoretical or data support should be provided for the advantages of the proposed new architecture. The feasibility and reliability of virtual machine management scheme are not convincing enough. Aspects of format and contextual consistency:

1) The content of line2 in the abstract is not consistent with that of line564 in conclusion.

2) The structural hierarchy specification of paragraphs. For instance, a paragraph number is needed for Line82, instead of a paragraph mark.

3) Chart format. For instance, there are overlapping and missing border in the table. There are intersected lines and text, inconsistent Title format and inappropriate text size in the picture.                                                                                   

Author Response

Find here attached  cover letter to explain point-by-point the
details of the revisions in the manuscript and our responses to the
reviewer 3' comments.

Round 2

Reviewer 2 Report

The authors seem to have modified the manuscript according to the suggestions.

Author Response

Find here attached.

Thanks

Reviewer 3 Report

Improvements have been made in the revised version. However, there are still some imperfections such as words or phrases in line132 and line222. Thus, this manuscript needs to be edited by a native English-speaking specialist. I hope these suggestions would be useful for improving the quality of the paper.

Author Response

Please find here attached

Thanks

This manuscript is a resubmission of an earlier submission. The following is a list of the peer review reports and author responses from that submission.

Round 1

Reviewer 1 Report

This paper provides a Software-defined vehicular cloud network which leverages SDN to address migration management of VM that supports cloud-based vehicular service. The cloud-based vehicle applications supported in SDVC are also introduced. However, there is no analysis of the feasibility of the architecture in this paper.

Furthermore, some detailed comments are as follows:

This paper uses cloud computing and SDN to improve connectivity and enhance bandwidth. How to prove the feasibility and effectiveness of the architecture. It would be better to add a comparative experiment. The proposed architecture includes three SDN controllers. Is there a central controller that controls these three controllers? In a MEC cloud, how do adjacent controllers collaborate to make decisions. In this paper, the name of the architecture is confusing, such as SDVC, SVCN. Please keep the name consistent. Try to keep the style of the picture consistent. And the figures are not standardized. For example, figure 5. There are some word errors that need to be corrected.

Reviewer 2 Report

The authors in this paper have presented an  SDN for Vehicular Cloud Networks, specifically on migration management of VM. The model and results collected show NO new contribution. The paper originality is low.

Poor network model scenario
1. You need results on complexity!
2. Hard Compare with other non-similar environments?!
3. Compare your work with more recent models such:
4. Proofread the paper.

Round 2

Reviewer 1 Report

This paper provides a Software-defined vehicular cloud network which leverages SDN to address migration management of VM that supports cloud-based vehicular service. However, the paper innovation and originality are low.

The feasibility of the scheme cannot be proven. Is the optimum alporithm mentioned in chapter 3 existing or new? Check the format.

Reviewer 2 Report

The authors have not answered my comments properly. It's like asking about a thing and they answer something else. The problem of the paper as follows:

New results neede to evaluate the feasibility of the contributions. Not added! No comparison with other work has been conducted in both versions. A discussion of existing work in the area is not enough.